Measurement invariance of the distress tolerance scale among university students with and without a history of non-suicidal self-injury

Slabbert Ashley
http://orcid.org/0000-0002-0172-9288 Hasking Penelope
Greene Danyelle
http://orcid.org/0000-0001-5420-8606 Boyes Mark mark.boyes@curtin.edu.au
School of Psychology, Curtin University , Perth, Western Australia , Australia
Jorm Anthony
Electronic publication date: 2021 Mar 15
Publication date: 2021
Volume: 9
Electronic Location ID: e10915
Received 2020 Oct 13; Accepted 2021 Jan 18
Copyright: © 2021 Slabbert et al.
Copyright year: 2021
Copyright holder: Slabbert et al.
License: This is an open access article distributed under the terms of the Creative Commons Attribution License, which permits unrestricted use, distribution, reproduction and adaptation in any medium and for any purpose provided that it is properly attributed. For attribution, the original author(s), title, publication source (PeerJ) and either DOI or URL of the article must be cited.
License URL: https://creativecommons.org/licenses/by/4.0/

Keywords: Non-suicidal self-injury, Distress tolerance, Measurement, Measurement invariance

Funding: National Health and Medical Research Council, Australia (Investigator) 1173043 Australian Government Research Training Program (RTP) Mark Boyes is supported by the National Health and Medical Research Council, Australia (Investigator Grant 1173043). Ashley Slabbert and Danyelle Greene are supported by Australian Government Research Training Program (RTP) scholarships. The funders had no role in study design, data collection and analysis, decision to publish, or preparation of the manuscript.

==============================
Non-suicidal self-injury (NSSI) is the intentional damage to one’s body tissue in the absence of suicidal intent. NSSI primarily serves an emotion regulation function, with individuals engaging in self-injury to escape intense or unwanted emotion. Low distress tolerance has been identified as a mechanism that underlies self-injury, and is commonly assessed using the self-report Distress Tolerance Scale. There are mixed findings regarding the factor structure of the Distress Tolerance Scale, with some researchers utilising a higher-order distress tolerance score (derived from the scores on the four lower-order subscales) and other researchers using the four subscales as unique predictors of psychological outcomes. Neither of these factor structures have been assessed among individuals with a history of self-injury. Of note, an inability to tolerate distress (thought to underlie NSSI) may limit an individual’s capacity to accurately observe and report specific thoughts and emotions experienced in a state of heightened distress, which may impact the validity of scores on the Distress Tolerance Scale. Therefore, measurement invariance should be established before attributing NSSI-related differences on the scale to true differences in distress tolerance. We compared the Distress Tolerance Scale higher-order model with the lower-order four factor model among university students with and without a history of NSSI. Our results indicated that the lower-order four factor model was a significantly better fit to the data than the higher-order model. We then tested the measurement invariance of this lower-order factor model among individuals with and without a history of NSSI, and established configural and full metric invariance, followed by partial scalar and full residual error invariance. These results suggest the four subscales of the Distress Tolerance Scale can be used to confidently discern NSSI-related differences in distress tolerance.

Introduction

Non-suicidal self-injury (NSSI) is the intentional damage to one’s body tissue in the absence of suicidal intent, for reasons not socially or culturally sanctioned (International Society for the Study of Self-Injury, 2018). International prevalence rates indicate approximately 13.4% of young adults report a history of self-injury, with elevated rates (20%) reported by university students (Swannell et al., 2014). NSSI is a behaviour receiving increasing attention from both researchers and clinicians, given its associations with negative psychological outcomes and heightened risk of suicide over time (Whitlock et al., 2013). Whilst there are various reasons for engaging in NSSI, individuals primarily report engaging in self-injury for emotion regulation purposes (Taylor et al., 2018). Several key theoretical models of NSSI, including the Emotional Cascade Model (Selby, Anestis & Joiner, 2008), the Experiential Avoidance Model (Chapman, Gratz & Brown, 2006) and the Cognitive-Emotional Model (Hasking et al., 2017), specify a central role for emotion regulation in the onset and maintenance of self-injury. According to these models and previous empirical research, heightened negative affect (Armey, Crowther & Miller, 2011; Boyes, Wilmot & Hasking, 2019; Najmi, Wegner & Nock, 2007; Slabbert et al., 2020), low positive affect (Bresin, 2014; Slabbert et al., 2020; Victor & Klonsky, 2014), greater repetitive negative thinking (Gong et al., 2019; Slabbert, Hasking & Boyes, 2018), as well as greater difficulties in emotion regulation (Gratz, Breetz & Tull, 2010; Jenkins & Schmitz, 2012) are all associated with increased likelihood of engaging in NSSI. Also common to these models is one’s ability to tolerate distress arising from emotional experiences.

Distress tolerance refers to both an individual’s perceived and actual ability to tolerate aversive physical and emotional states (Leyro, Zvolensky & Bernstein, 2010). Theoretically, individuals who experience greater difficulties tolerating intense emotion are less willing (or able) to withstand distress and more likely to self-injure as a means of escaping the aversive emotional state (Chapman, Gratz & Brown, 2006). Researchers have established direct links between low distress tolerance and NSSI; individuals with lower levels of distress tolerance are more likely to report a history of self-injury (Anestis et al., 2013; Lin et al., 2018; Slabbert, Hasking & Boyes, 2018) as well as more frequent NSSI (Anestis et al., 2013). Distress tolerance is typically assessed with self-report measures, most commonly the Distress Tolerance Scale (Simons & Gaher, 2005).

The Distress Tolerance Scale (Simons & Gaher, 2005) is a multidimensional scale designed to capture four core facets of distress tolerance; an individual’s perceived ability to tolerate emotional distress (tolerance), subjective appraisal of distress regarding whether the distress is seen as acceptable or shameful (appraisal), the level of attention absorbed by distressing emotions (absorption), and efforts taken to alleviate the distress (regulation). A higher-order global distress tolerance score is derived by averaging the scores on the four subscales. The internal consistency of the higher-order scale and lower-order scales is generally good, with convergent and divergent validity previously established (Leyro et al., 2010; Simons & Gaher, 2005).

There is evidence to support the higher-order factor structure (Leyro et al., 2010; Sandín et al., 2017; Werner-Seidler et al., 2013), and many researchers opt to only utilise the total distress tolerance score in their research (Anestis et al., 2013; Hovrud et al., 2019; Peterson, Davis-Becker & Fischer, 2014). However, there is growing acknowledgment that the tendency to only focus on global distress tolerance has resulted in researchers losing potentially important information captured in the individual subscales that may better explain relationships between distress tolerance and psychopathology, or behaviours such as NSSI (Leyro et al., 2010). In studies where researchers have elected to investigate the four subscale scores, findings indicate that some subscales may be more salient than others in predicting psychopathology such as anxiety and depression, as well as dysregulated behaviours including self-injury (Horgan & Martin, 2016; Lin et al., 2018).

The two different factor structures have only been directly compared in one study. Among a sample of Chinese adolescents, You & Leung (2012) found both the higher-order factor model and lower-order four factor model demonstrated better fit than 1-factor and 2-factor structures with which they were compared, with the higher-order model demonstrating best fit. Despite You & Leung’s (2012) findings, there is some evidence to suggest lower-order factor models may demonstrate better model fit than higher-order models (Meganck, Vanheule & Desmet, 2008). Therefore, it is necessary to evaluate the fit of each of these models among individuals with a history of NSSI to further our theoretical understanding of the nature of the relationship between distress tolerance and self-injury, and consequently inform researchers about the optimal way to utilise this scale with samples of individuals with a history of NSSI.

Another growing concern regarding the measurement of constructs such as distress tolerance, is the accuracy of the heavily relied upon self-report scales such as the Distress Tolerance Scale to detect true group differences. Researchers have become increasingly aware that statistically observed differences on these scales are only meaningful when these instruments demonstrate invariance across groups (Sass, 2016). Measurement noninvariance may have several problematic implications. For example using non-invariant scales to assess the severity of psychological disorders such as depression across groups (i.e. women and men) may result in one group (i.e. men) scoring lower than the other, simply because they interpret the items differently, as opposed to actually experiencing less severe depression (Putnick & Bornstein, 2016). Consequently, our understanding based on these findings is that females experience more severe depression than men which may not be accurate, but directs future research towards female-oriented studies and interventions (Putnick & Bornstein, 2016). Relatedly, another example of where measurement invariance is problematic is the use of pre-test and post-test measurements to assess the effectiveness of an intervention or clinical trial. It is possible that the intervention or trial itself may impact how participants interpret the constructs being assessed (Putnick & Bornstein, 2016). Consequently, this may result in inaccurate conclusions regarding the effectiveness of an intervention (Putnick & Bornstein, 2016). These example highlight the importance of establishing measurement invariance in psychological science.

Recent research testing the measurement invariance of three emotion regulation questionnaires in young adults with and without a history of NSSI showed that observed NSSI-related differences on the Difficulties in Emotion Regulation Scale—Short Form (DERS-SF; Gratz & Roemer, 2004) and the Cognitive Emotion Regulation Questionnaire—Short (CERQ-S; Garnefski & Kraaij, 2007) were reliable and likely a true reflection of group differences in emotion regulation (Kiekens, Hasking & Boyes, 2019). However, the widely used Emotion Regulation Questionnaire (ERQ; Gross & John, 2003) did not demonstrate measurement invariance, with two of the items on the Cognitive Reappraisal subscale functioning differently for individuals with a history of self-injury compared to individuals who had never self-injured. This is concerning as previous research that has established NSSI-related differences in cognitive reappraisal using this scale may be reflecting a measurement artefact rather than true group differences. Similarly, Greene et al. (2020) established that the Externally Oriented Thinking subscale of the frequently used Toronto Alexithymia Scale (TAS-20; Bagby et al., 2006) was not invariant across individuals with and without a history of NSSI, precluding any conclusions regarding NSSI-related differences in externally oriented thinking. These findings highlight the importance of investigating the measurement invariance of self-report measures to reveal whether or not we can reliably draw conclusions about particular group differences using these assessment tools.

Measurement invariance of the Distress Tolerance Scale between individuals with and without a history of NSSI has not yet been assessed. It is plausible that the very difficulties in withstanding distress underlying dysregulated behaviours such as NSSI, may limit an individual’s capacity to accurately observe and report specific thoughts and emotions experienced in a heightened distressed state, which may impact the validity of results on self-report measures such as the Distress Tolerance Scale. Observed differences in distress tolerance between people who do and do not self-injure may be a function of a differential interpretation of scale items, rather than a reflection of true group differences in distress tolerance. Given results derived from these measures are used by researchers and clinicians to inform future prevention and intervention programs, it is vital that we ensure these instruments are able to accurately produce reliable results across individuals with and without a history of self-injury. Additionally, this may have implications for the existing body of literature that has established NSSI-related differences in self-report distress tolerance using this scale, and future researchers may need to be cautious when using these findings to justify their aims or results.

This study had two primary aims. First, to test and compare the higher-order model of the Distress Tolerance Scale to the lower-order four factor model to determine the best fitting model among a sample of university students, as well as within subgroups of individuals with and without a history of NSSI. Second, to test measurement invariance of the best fitting model between individuals with and without a history of self-injury.

Materials and Methods

Participants and procedure

Total sample

Participants were 531 Australian University students (74.7% female) between the ages of 17 and 25 (M = 20.58, SD = 1.94) recruited through an undergraduate participant pool and social media platforms. Of participants, 412 (77.6%) were born in Australia, followed by India (2.6%) and Malaysia (2.3%). The majority of participants were currently completing an undergraduate bachelor degree (96%), followed by a Master degree (2.4%). In total, 171 (32.2%) of individuals reported a history of mental illness, most commonly anxiety and depression.

History of NSSI

Of the total sample, 215 individuals who reported a prior history of NSSI (Mage = 20.87, SD = 2.0). Of these, 188 (87.4%) were female, 173 (80.5%) were born in Australia, and 202 (94%) were studying an undergraduate bachelor degree. With regards to mental illness, 127 (59.1%) participants reported a history of mental illness, most commonly anxiety and depression.

No history of NSSI

Of the total sample, 316 participants reported never engaging in NSSI (Mage = 20.38, SD = 1.88). Of these, 209 were female (66.1%), 239 (75.6%) were born in Australia, and 308 (97.5%) were studying an undergraduate bachelor degree. With regards to mental illness, 44 (13.9%) participants reported a history of mental illness, again most commonly anxiety and depression.

After providing informed consent, participants completed a series of online questionnaires hosted by Qualtrics. Data were collected as part of a larger study investigating the role of social, cognitive, and emotional factors underlying health risk behaviours. Students received either course credit or were entered into a prize draw to win an iPad or $50 gift cards. This study received ethical approval from the Curtin University. The study received ethical approval from the Curtin University Human Research Ethics Committee (HRE2018-0536) and participants were provided with a list of counselling resources and information about self-injury upon completion of the survey.

Measures

Distress tolerance

The 15-item Distress Tolerance Scale (Simons & Gaher, 2005) was used to assess individual differences in the ability to experience and withstand negative psychological states. Items are rated on a 5-point Likert scale (1: strongly agree; 5: strongly disagree), with higher scores reflecting higher levels of distress tolerance. The scale consists of four subscales: tolerance (three items, for example ‘I can’t handle feeling distressed or upset’), appraisal (six items, for example ‘My feelings of distress or being upset are not acceptable’), absorption (three items, for example ‘My feelings of distress are so intense that they completely take over’) and regulation (three items, for example ‘I’ll do anything to avoid feeling distressed or upset’). Subscale scores are calculated by averaging response to all items on each subscale. A higher-order distress tolerance score is calculated by averaging the subscale mean scores. This scale demonstrates excellent internal consistency (α = 0.89; Simons & Gaher, 2005). Internal consistencies were adequate to excellent in the current sample (Global Score, α = 0.93, ω = 0.93; Tolerance, α = 0.84, ω = 0.85; Appraisal, α = 0.85, ω = 0.86; Absorption α = 0.86, ω = 0.86; Regulation, α = 0.76, ω = 0.79).

Non-suicidal self-injury

Section I of the Inventory of Statements About Self-Injury (ISAS; Klonsky & Glenn, 2008) was used to assess history and frequency of NSSI (defined to participants as intentionally harming oneself without intention to suicide). Individuals were first provided with a definition of NSSI and then asked ‘Have you ever engaged in non-suicidal self-injury?.’ Participant who responded yes to this question were then asked to report a lifetime frequency of twelve common methods of NSSI (e.g. cutting, scratching, burning). The ISAS demonstrates good four week test-retest reliability, (r = 0.85; Klonsky & Olino, 2008).

Data analysis

To determine the best fitting model, we examined the model fit of the original higher-order factor structure and the lower-order four factor structure of the Distress Tolerance Scale using a sequence of Confirmatory Factor Analyses with a Maximum Likelihood Estimation with robust standard errors and a mean- and variance adjusted test statistic (MLMV). These analyses were conducted among the total sample, the sub-sample of individuals with a history of NSSI, and the sub-sample of individuals without a history of NSSI. A model demonstrated acceptable fit if it met the following criteria: Comparative Fit Index (CFI) and Tucker-Lewis Index (TLI) values between 0.90 (adequate) and 0.95 (good) or higher, and a Standardized Root Mean Square Residual (SRMR) and Root Mean Square Error of Approximation (RMSEA) values close to or below 0.08 (Brown, 2015). Modification indices suggested residual variances be correlated to improve model fit. Given there was a cluster of items with error covariances above 0.40 on the same subscale, we had theoretical justification for allowing these items to correlate in order to improve model fit (Whittaker, 2012). A chi-square difference test was conducted to statistically compare the two models.

We then tested for measurement invariance across individuals with and without a history of NSSI using the best-fitting model using a multigroup confirmatory factor analysis (MGCFA) with Maximum Likelihood Estimation with robust standard errors and a mean- and variance adjusted test statistic (MLMV). We assessed configural (i.e. equal pattern of factor loadings), metric (i.e. equal factor loadings), scalar (i.e. equal factor loadings and equal intercepts), and residual error invariance (i.e. equal factor loadings, equal intercepts, and equal residual error variance uniqueness). Measurement invariance was supported if the configural model demonstrated acceptable fit and each of the subsequent models showed a non-significant change in chi-square test statistic and a change in CFI of <0.01 and in RMSEA of <0.015 and SRMR of <0.030 (for metric invariance) or <0.015 (for scalar or residual invariance; Chen, 2007) from the previous levels. Partial invariance will be addressed using a sequential backwards approach where items are freed until partial invariance is achieved (Putnick & Bornstein, 2016). All analyses were conducted using MPlus v7.4 (Muthen & Muthen, 2017)1 .

Results

Results from a Missing Values Analysis indicated data were missing not completely at random, χ2(4,012) = 4,204.185, p = 0.02, however given less than 5% of data were missing on all variables, Expectation Maximization was used to impute missing data (Tabachnick & Fidell, 2013). Of the 531 participants, 215 (40.5%) reported a history of NSSI, with 118 (54.9%) of these individuals reporting engaging in self-injury in the past 12 months. The primary method of NSSI was cutting (50%), followed by severe scratching (12.4%), and self-battery (11.4%). Age of onset ranged from 4 to 23 years (M = 13.69, SD = 2.99). Descriptive statistics are presented in Table 1.

Table 1 Descriptive statistics disaggregated by history of NSSI.

	Total sample	No history of NSSI (n = 316)	History of NSSI (n = 215)	ta		
	M (SD)	Skewness (SD)	Kurtosis (SD)	M (SD)	Skewness (SD)	Kurtosis (SD)	M (SD)	Skewness (SD)	Kurtosis (SD)		
Tolerance	2.94 (1.07)	0.11 (0.11)	−0.64 (0.21)	3.22 (1.01)	0.04 (0.14)	−0.66 (0.27)	2.53 (1.01)	0.31 (0.17)	−0.48 (0.33)	7.70***	
Appraisal	3.09 (0.94)	0.03 (0.11)	−0.61 (0.21)	3.42 (0.84)	0.02 (0.14)	−0.62 (0.27)	2.59 (0.87)	0.29 (0.17)	−0.52 (0.33)	11.00***	
Absorption	2.81 (1.10)	0.18 (0.11)	−0.76 (0.21)	3.18 (1.01)	0.02 (0.14)	−0.67 (0.27)	2.25 (0.98)	0.56 (0.17)	−0.37 (0.33)	10.63***	
Regulation	2.90 (.92)	0.11 (0.11)	−0.27 (0.21)	3.03 (0.89)	0.16 (0.14)	−0.11 (0.27)	2.70 (0.94)	0.14 (0.17)	−0.50 (0.33)	4.12***	
Total DTS score	2.93 (.85)	0.07 (0.11)	−0.35 (0.21)	3.22 (0.79)	0.04 (0.14)	−0.29 (0.27)	2.52 (0.77)	0.13 (0.17)	−0.42 (0.33)	10.14***	
Notes:

a t Values are in reference to the mean comparison between individuals with and without a history of NSSI.

*** p < 0.001.

Factor structure evaluation

Both the original higher-order model and the lower-order four factor model demonstrated adequate baseline fit in the total sample and among individuals without a history of NSSI, but demonstrated poorer fit among individuals with a history of NSSI (Table 2). Given item 7 ‘My feelings of distress or being upset are not acceptable’ and 11 ‘I am ashamed of myself when I feel distressed or upset’ were both on the appraisal subscale and had error covariance larger than 0.40, these two items were allowed to correlate. Item 11 and 12 ‘My feelings of distress or being upset scare me’ were also on the appraisal subscale and had error variance larger than 0.40 so were allowed to correlate.

Table 2 Comparison of DTS models among the total sample, individuals with a history of NSSI, and individuals without a history of NSSI.

	χ2	df	Δ χ2 (Δ df)	p Δ χ2	CFI	TLI	RMSEA	SRMR	
Total sample (n = 531)	
Baseline fit		
 Higher-order factor model	356.534	86	–	–	0.915	0.896	0.077	0.059	
 Lower-order factor model	312.991	82	43.543 (2)	<0.001	0.928	0.910	0.072	0.051	
Baseline fit with appraisal item correlations	
 Higher-order factor model	280.973	84	–	–	0.938	0.923	0.066	0.055	
 Lower-order factor model	245.490	82	35.483 (2)	<0.001	0.949	0.934	0.061	.047	
 NSSI history (n = 215)	
Baseline fit	
 Higher-order factor model	210.779	86	–		0.872	0.844	0.082	0.087	
 Lower-order factor model	181.950	84	28.829(2)	<0.001	0.899	0.874	0.074	0.071	
Baseline fit with appraisal item correlations	
 Higher-order factor model	186.424	84	–	–	0.895	0.869	0.075	0.084	
 Lower-order factor model	159.979	82	26.45 (2)	<0.001	0.920	0.897	0.067	0.067	
 No NSSI history (n = 316)	
Baseline fit	
 Higher-order factor model	200.891	86	–		0.923	0.906	0.065	0.053	
 Lower-order factor model	188.532	84	12.359 (2)	0.002	0.930	0.913	0.063	0.051	
Baseline fit with appraisal item correlations	
 Higher-order factor model	153.183	84	–	–	0.954	0.942	0.051	0.049	
 Lower-order factor model	145.198	82	7.99 (2)	0.018	0.958	0.946	0.049	0.045	

Results from chi-square difference tests indicate allowing these items to correlate significantly improved the baseline fit of the higher-order model in the total sample Δχ2(2) = 75.561, p < 0.001, within the sub-sample of individuals with a history of NSSI Δχ2(2) = 24.355, p < 0.001, and within the sub-sample of individuals without a history of NSSI Δχ2(2) = 47.708, p < 0.001. Despite this improvement, the higher-order model still remained a poor fit among individuals with a history of NSSI (Table 2).

Comparatively, these modifications resulted in the lower-order four factor model demonstrating good fit within the total sample Δχ2(2) = 67.501, p < 0.001, and among individuals without a history of NSSI Δχ2 (2) = 43.334, p < 0.001 and adequate fit among individuals with a history of NSSI, Δχ2 (2) = 21.971, p < 0.001. Importantly, chi-square difference tests indicated the lower-order four factor model was a significantly better fit than the higher-order model in all three groups (Table 2). The factor loadings (full sample) for both the higher-order and lower-order factor models are illustrated in Figs. 1 and 2.

Figure 1 Distress tolerance scale higher-order factor model (total sample).

Figure 2 Distress tolerance scale lower-order four factor model (total sample).

Measurement invariance of the DTS lower order 4 factor model

Given the correlated lower-order four factor model was the best fit, this is the model we chose to evaluate for measurement invariance. Configural (M1) and full metric (M2) invariance was supported for the lower-order four factor model, but the Δχ2 test statistic indicated full scalar (M3.1) invariance was not supported (Table 3). To address achieve partial invariance, we identified the source of non-invariance by sequentially releasing item intercept constraints until the model was invariant (Putnick & Bornstein, 2016). We identified that releasing item 10 intercept constraints had the most influential impact on model fit. Irrespective of the score on the underlying latent factor appraisal, there was a tendency for young adults who self-injured to agree more with item 10 ‘Being distressed or upset is always a major ordeal for me’ (Intercept(No NSSI) = 3.31 vs. Intercept(NSSI) = 2.56). Allowing these intercepts to vary between groups, partial scalar (M3.2) and full residual error (M4) invariance was supported.

Table 3 Measurement invariance of the lower-order four factor distress tolerance scale.

	χ2	df	Δ χ2 (Δ df)	p Δ χ2	NCI	CFI	RMSEA	SRMR	Model comparison	ΔNCI	ΔCFI	ΔRMSEA	ΔSRMR	
Model 1: configural invariance	304.965	164	–	–	0.8755	0.943	0.057	0.055		–	–	–	–	
Model 2: full metric invariance	315.623	175	10.66 (11)	0.472	0.8758	0.943	0.055	0.058	M1–M2	0.0003	<0.001	0.002	0.003	
Model 3.1: full scalar invariance	338.147	186	22.52 (11)	0.021	0.8663	0.939	0.056	0.063	M2–M3.1	0.0100	0.004	0.001	0.005	
Model 3.2: partial scalar invariancea	328.614	185	12.99 (10)	0.224	0.8733	0.942	0.054	0.060	M2–M3.2	0.0035	0.001	0.001	0.002	
Model 4: full residual error invariance	348.770	200	20.16 (15)	0.166	0.8691	0.940	0.053	0.062	M3.2–M4	0.0042	0.002	0.001	0.002	
Note:

a Intercept of item 10 (p < 0.001) was lower in people who self-injure.

There were significant latent mean differences, with individuals with a history of NSSI scoring lower than those with no history on the tolerance subscale (Z = −7.92, p < 0.001), absorption subscale (Z = −10.20, p < 0.001), and regulation subscale (Z = −4.52, p < 0.001). Regardless of whether the differential item functioning of item 10 on the appraisal subscale was considered (Z = −8.08, p < 0.001), or ignored (Z = −7.68, p < 0.001), individuals with a history of NSSI scored lower than individuals without a history of NSSI.

Discussion

Many studies use the self-report Distress Tolerance Scale to examine group differences in distress tolerance between individuals with and without a history of self-injury (Anestis et al., 2013; Horgan & Martin, 2016; Lin et al., 2018; Slabbert, Hasking & Boyes, 2018). Relative to individuals with no history of self-injury, individuals with a history of NSSI report less global distress tolerance (Anestis et al., 2013; Lin et al., 2018; Slabbert, Hasking & Boyes, 2018), with group differences specifically observed on the appraisal and absorption subscales (Horgan & Martin, 2016; Slabbert et al., 2020). However, to ensure confidence in these findings it is important that we confirm the psychometric properties, including measurement invariance, of the Distress Tolerance Scale among individuals with and without a history of self-injury. The aim of the current study was to compare the higher-order and the lower-order four factor structure of the scale among a sample of university students, as well as within the sub-samples of individuals with and without a history of NSSI, to determine the best fitting model, and whether this was invariant across individuals with and without a history of NSSI.

Results indicated that the lower-order four factor structure demonstrated superior fit in all analyses. Based on these findings, measurement invariance analyses were conducted on the lower-order four factor model, with results indicating full invariance at the configural and metric level, followed by partial scalar and full residual error invariance. Despite freeing one item intercept at the scalar level, observation of latent mean differences indicate that all subscales can be confidently used to assess NSSI-related group differences in distress tolerance.

The findings suggest that using the four subscales of the Distress Tolerance Scale including Tolerance, Appraisal, Absorption and Regulation, as unique predictors of outcomes such as NSSI, as opposed to a single distress tolerance score, may be statistically superior. It is not uncommon for lower-order factor models to demonstrate better fit than when the lower-order factors are forced to load onto a higher-order factor, with similar results evident in self-report measurement of alexithymia (Meganck, Vanheule & Desmet, 2008). However, it is important to acknowledge that the difference between these two models, although significant, was not large. With some minor modifications, the higher-order factor model still demonstrated adequate fit in the full sample and among individuals with no history of self-injury. This higher-order model may prove useful in research contexts where a global distress tolerance score is valuable, perhaps in studies where researchers are interested in a broad range of constructs and require a more simplistic and direct way of assessing distress tolerance.

However, the use of the individual subscales may provide a more comprehensive and holistic understanding of key elements underlying distress tolerance and its relationship with psychopathology and behaviours such as NSSI. For example in one of the few studies that examined associations between the four distress tolerance subscales and NSSI, Horgan & Martin (2016) established that only the appraisal and absorption subscales differentiated people with and without a history of self-injury. Similarly, Slabbert et al. (2020) found that the appraisal and absorption scales differentiated between individuals who had recently self-injured and individuals who had never self-injured. Their results also indicated that experiencing greater positive affect might protect against negative appraisals of distress. Based on these findings, how one views their distress and how much attention they allocate towards this distress, appear to be more important in predicting NSSI than an individual’s perception of the their tolerance or how they attempt to regulate their distress. Employing this four factor model in future research will allows researchers to delve deeper into the relationship between distress tolerance and self-injury and consequently gain a more accurate and nuanced understanding regarding how different aspects of distress tolerance are related to NSSI.

After establishing that the lower-order four factor model was the superior fitting model, we investigated whether it was invariant among individuals with and without a history of NSSI. Our results were promising, with full configural and metric invariance being supported, and after freeing the item intercept for one item (“Being distressed or upset is always a major deal for me”), partial scalar invariance was achieved. Consequently, full residual error invariance was satisfied however this was contingent on the partial scalar model where the intercept constraints for item 10 were released. When examining this item, it does not appear to differ in terms of its content in comparison to other items, such that it fits well within the general concept of perceived tolerance of distress tolerance. If it appeared to assess something more abstract or obscure in comparison to the other items this may explain differences in interpretation however this does not appear to be the case. Another factor thought to impact interpretation of items is whether they are positively or negatively keyed (Meganck, Vanheule & Desmet, 2008). Previous research has established noninvariance between groups on negatively keyed items (Lindwall et al., 2012), however once again this is not the case with regards to item 10 on the Distress Tolerance Scale as it is positively keyed. Therefore it is not clear why individuals with a history of NSSI may have a different interpretation of this item compared to individuals who have never self-injured. However, the strictest test of invariance was employed in this analysis (chi-square difference test). If we had employed the more liberal criteria used to assess measurement invariance which supports invariance if the difference in CFI between the configural level and other levels is less than 0.01 (Chen, 2007) then full scalar invariance would have been achieved. Regardless, whether this item intercept was freed or not, there were still significant mean differences on the appraisal subscale with individuals with a history of NSSI tending to appraise their distress as more unacceptable than individuals with no history of NSSI. These results instil confidence that we are able to reliably detect real group differences in distress tolerance between individuals with and without a history of NSSI using this four-factor Distress Tolerance Scale.

Whilst the findings of this study provide promising support for the use of the Distress Tolerance Scale to examine NSSI-related group differences in distress tolerance, there are several limitations that warrant consideration. The sample predominantly comprised female university students who self-selected into the study, meaning these findings may not generalise to a community sample. Additionally, while NSSI is prevalent amongst university students, it is unlikely that many would meet the diagnostic criteria for the proposed NSSI disorder (Kiekens, Hasking & Boyes, 2019). Individuals who meet this criteria would likely experience significantly greater difficulties in emotion regulation and consequently may also have more difficulty reflecting on previous times of heightened distress and reporting on their ability to tolerate distress. Therefore, investigation of measurement invariance of the Distress Tolerance Scale in clinical samples is warranted.

Conclusion

In evaluating the two models of the Distress Tolerance Scale, as well as testing the measurement invariance of the lower-order four factor model, this study has provided support for the use of this scale to reliably assess NSSI-related group differences. The lower-order four factor model appears to be statistically superior and may offer a more comprehensive understanding of the relationship between the specific facets of distress tolerance and behaviours such as NSSI. Additionally, the lower-order four factor model demonstrated invariance up until the scalar level according to the strictest invariance criteria, requiring only one item intercept to be freed to satisfy partial and full residual error invariance. Although further investigation in other samples is required, these results suggest that the Distress Tolerance Scale can be used with confidence that true group differences will be reflected in scores.

Supplemental Information

Supplemental Information 1 Distress Tolerance Scale item means and standard deviations for the total sample, individuals with a history of NSSI, and individuals without a history of NSSI.

Click here for additional data file.

Supplemental Information 2 Measurement invariance of the first-order four factor Distress Tolerance Scale between males and females.

Click here for additional data file.

Supplemental Information 3 DTS NSSI-related measurement invariance data.

Click here for additional data file.

Supplemental Information 4 Dataset Codebook.

Click here for additional data file.

Additional Information and Declarations

Competing Interests

Author Contributions

Human Ethics

Data Availability

1 Given the majority of participants were female, and at the request of an anonymous reviewer, we also tested measurement invariance across gender. Results indicated that configural, metric, scalar, and residual error invariance across gender were all supported, and thus we conclude males and females do not respond differently to items on this scale (see Table S2).

Mark Boyes is an Academic Editor for PeerJ.

Ashley Slabbert conceived and designed the experiments, performed the experiments, analyzed the data, prepared figures and/or tables, authored or reviewed drafts of the paper, and approved the final draft.

Penelope Hasking conceived and designed the experiments, authored or reviewed drafts of the paper, and approved the final draft.

Danyelle Greene performed the experiments, analyzed the data, authored or reviewed drafts of the paper, and approved the final draft.

Mark Boyes conceived and designed the experiments, authored or reviewed drafts of the paper, and approved the final draft.

The following information was supplied relating to ethical approvals (i.e., approving body and any reference numbers):

The Curtin University Human Research Ethics Committee granted ethical approval for the study (HRE2018-0536).

The following information was supplied regarding data availability:

Raw data, including data on NSSI history, responses to the 15 DTS items, scores for the four DTS subscales, and total DTS score, are available in the Supplemental Files.

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
