# Peer review of "Measurement invariance of the distress tolerance scale among university students with and without a history of non-suicidal self-injury"

_PeerJ, doi:10.7717/peerj.10915_

## Round 0.1 · original submission · Major Revisions

In addition to the reviewers' comments below, I note that you use Cronbach's alpha as a measure of reliability. Psychometricians nowadays generally recommend use of omega over alpha. See for example https://onlinelibrary.wiley.com/doi/abs/10.1111/bjop.12046.

Reviewer 1 ·

Basic reporting

The authors conducted a test of measurement invariance of the Distress Tolerance Scale across status of non-suicidal self-injury among university students in Australia. Overall, the paper is well-organized and the writing is clear throughout. I just have a few comments for the authors:

Introduction
1. Was it absolutely necessary to discuss measurement invariance findings on self-report measures of alexithymia (lines 141-154)? Readers might feel a little distracted from the main flow of the discussion on existing literature.
2. I understand that the single focus of this paper was on measurement invariance of the Distress Tolerance Scale, however, it might be more enlightening to offer a little background on other significant correlates of non-suicidal self-injury in the Introduction section or Methods section, such as depression, distress, etc. However, I would like to hear the authors’ arguments on not including or discussing other significant correlates of NSSI.

Experimental design

1. Can the authors provide explanations as to why there was minimum discussion on demographic information of the sample such as academic performance, annual family income, or lifestyle factors? It would be nice to include such information in Table 1.
2. Majority (74.7%) of the sample were females, which might be of concern as to if findings on measurement invariance of the DTS would apply more to females rather than males?
3. It would also be nice to include information on the normality indices of the sample such as skewness and kurtosis, as well as information on internal reliability coefficient of the DTS.
4. Factor loadings of item 6 and item 14 seemed a bit low.

Validity of the findings

Sufficient discussion on findings and limitations.

Additional comments

no further comment to add.

Reviewer 2 ·

Basic reporting

a. Overall, the current manuscript used clear, grammatically correct English. Still, some clarification is warranted as detailed in the Validity of the Findings.
b. The manuscript seems to include relevant prior literature.
c. The structure of the manuscript appears appropriate.
d. Raw data was shared.
e. The relevant results being reported appear appropriate based on the described analytic strategies.

Experimental design

a. Research question appears to be clearly defined. Nonetheless, some clarification on the study rationale is warranted as detailed below under Validity of the Findings.
b. Methods warrant some clarifications including handling missing data and analytic strategies addressing partial measurement invariance (MI). Please see the details below under Validity of the Findings.

Validity of the findings

a. The finding is novel. However, the current conclusion warrants caution due to some missing information (e.g., investigation on the reasons for low Cronbach’s alpha, strategies to address partial MI), as detailed below.

Title
- Specifying the study sample would be helpful for readers. For example, authors could say ‘college students’ as opposed to ‘students’ to indicate that the sample was comprised of young adult individuals.

Introduction
- Line 125-154: These two paragraphs discuss potential issues arising when using scales without established MI among those with NSSI. While the authors did a nice job presenting specific examples, it is not clear what poses concerns in utilizing scales without established MI. Despite a brief summary of potential implications in lines 138-140 and 150-154, the current argument is limited. Why is using scales without established MI problematic above and beyond statistical concern? What are the consequences of using those scales in understanding NSSI? What are the clinical implications? Why is it crucial to establish MI in the DTS?
- Line 156: There is a typo in “NSIS”.

Materials and Methods
- Participants and procedure: Please provide information on race, ethnicity, and school year if such information is available. Was any information on the history of mental illness available? If so, please provide it. Also, please provide demographic information by group (i.e., NSSI vs no-NSSI).
- Line 204-205: Authors refer to two analytic models as “higher-order factor structure” and “first-order four-factor structure,” which do not correspond to the model labels in Table 2 and Figure 1. Please use consistent language.
- Line 213: Citation “Brown 2015” is missing in the reference.
- Line 215-226: The paragraph describes the procedure of MGCFA and the criteria. However, the information on the groups in comparison is missing.
- Line 220-221: What do you mean by “both models”?
- Overall, how was the missing data handled (e.g., the proportion of missing data, imputation method, the assumption underlying missing data)?

Results
- Line 234: The models are referred to as “higher-order model and the lower-order four-factor model” which do not correspond to the labels in Table 2, Figure 1, and previous sections. Please use consistent language.
- Line 237-238: Authors explain the modification indices. I would recommend explaining this procedure with appropriate citations in the Data Analysis section so that readers can understand why the authors made this decision.
- Line 264-266: In testing MI, scalar invariance was not supported and the items were investigated. Authors state that there was a response tendency to item 10 such that the NSSI group tended to respond by agreeing more. However, reviewing the shared data, a similar response tendency was found in items 2, 3, 4, 5, 9, 11, 12, and 15 among those with a history of NSSI. What was the rationale to pick item 10 only to vary? Related, the Cronbach’s alpha is very low for the Appraisal subscale (0.69) which is composed of items 9, 11, and 12 including item 10. Please provide a clear rationale/strategy in handling the partial scalar invariance in the Data Analysis section. For example, the following reference might help:
o Byrne, B. M., Shavelson, R. J., & Muthén, B. O. (1989). Testing for the equivalence of factor covariance and mean structures: The issue of partial measurement invariance. Psychological Bulletin, 105, 456–466. http://dx.doi.org/10.1037/0033-2909.105.3.456
- Lines 262-268: Authors state that “full metric and residual error” was supported. I am not sure if the term being used is following the conventional terms of MI research. Typically, the terms ‘strong’ or ‘weak’ MI are used as opposed to ‘full.’ I would recommend reconsidering using the term ‘full’ in the Results and Discussion section. Additionally, the residual error invariance in the current data is contingent on the partial scalar MI model. Authors may want to be cautious when reporting the results of residual error invariance.

Discussion
- In general, I would expect to see some discussion on the low Cronbach’s alpha in the Appraisal subscale, that might have contributed to partial scalar MI.
- The current conclusion should acknowledge the partial scalar invariance and the residual error invariance that is contingent on the partial scalar invariance.
- Line 318-320: This sentence carries important clinical implications in appraisal style among individuals with NSSI. However, related to my first comment on low Cronbach’s alpha in the Appraisal subscale, I recommend the authors to investigate the items in the Appraisal subscale at the analysis stage so that this sentence is supported.
- Line 328-330: This sentence is unclear. What do you mean by ‘in terms of its content in comparison to other items’? Can you expand more on what it means with ‘previously thought to influence interpretation’?
- Line 332-333: Authors state that they used the strictest test of invariance, referring to the chi-square difference test. What is the reference to support this sentence? In my understanding, the chi-square test is sensitive to sample size, which is the reason why other indices such as CFI, RMSEA, and SRMR are used according to Chen, 2007.

Tables and Figures
- Table 1: Readers would benefit from the descriptive information from the full sample. Please also double-check minor typos in the decimals in the column reporting t values as I noticed some typos when I ran the same analysis using the shared data.
- Table 2: Please use consistent language when labeling models (e.g., Second-order factor model or higher-order factor model as stated in the text body).
- Table 3: What does “M” stand for? When referring to each model, the term “Full” seems misleading.
- I would recommend having a supplemental table comparing the mean and SDs of each item in DTS in both the total sample and two subsamples.

Additional comments

The current manuscript examines the measurement invariance (MI; configural, metric, scalar, and residual error) of the 15-item Distress Tolerance Scale (DTS). The study sample was comprised of college students (N=531). The history of non-suicidal self-injury (NSSI) was self-reported. Two analytic approaches were implemented. First, two models (higher-order and first-order model) were tested among the total sample and two subsamples (history of NSSI vs no history of NSSI) in order to identify the best fitting model. Second, the first-order model was selected to test MI in the aforementioned two subsamples. Results supported configural and metric invariance. However, the results supported partial scalar invariance and residual error invariance contingent on the model of partial scalar invariance. The authors conclude that the 15-item DTS reflects group differences in NSSI. Together, this is a well-written manuscript with an important contribution in clarifying the potential measurement issue in DTS in assessing the level of distress tolerance among individuals with and without NSSI history. Nonetheless, some major methodological clarifications are warranted.

·

Basic reporting

The article “Measurement invariance of the Distress Tolerance Scale among students with and without a history of non-suicidal self-injury” (#53519) clearly and effectively describes the process of analysis of the factor structure of the aforementioned scale, and explores its measurement invariance among these groups of students. The literature review provides adequate background on the concepts relevant for the research and on the instruments used, showing conceptual accuracy and the pertinence of its aims. The article is very well structured and organized, respecting the standards proposed by the journal. It is also well written, with clear and technically accurate discourse that respects professional standards. The figures and tables are relevant, well-labelled and have enough quality. Raw data is supplied. Through this we can calculate the descriptive statistics of individual items to assess their kurtosis and skewness. However, it could be useful for readers to include this additional information in a table. Important results for the two aims proposed are presented and adequately discussed.

Experimental design

The research presented conforms to the journal’s aims and scope and it very clearly defines its aims. The article clearly argues how the research fills a knowledge gap in the area, clarifying the use of the Distress Tolerance Scale according to the factorial structure chosen by researchers. Therefore, it provides evidence for the potential of the scale’s use with a first-order four factor structure, since it may provide “a more comprehensive and holistic understanding of key elements underlying distress tolerance and its relationship with psychopathology and behaviours such as NSSI” (lines 309-311), avoiding simplistic interpretations of results. Moreover, it evidences satisfactory levels of measurement invariance for students with and without NSSI, which supports the psychometric properties and value of the scale and its potential usefulness for further understanding these behaviours. However, the strategy used to achieve full residual error invariance could be considered by some to be debatable. Indeed, this level of measurement invariance is not common in psychological measures and results could be considered sufficient without resorting to this strategy. Ethical standards seem to have been respected since ethical approval was obtained, authors present how participant consent was obtained, and participants received information about self-injury upon completion of the survey (line 181). The article provides adequate information on the statistical analyses performed, showing high technical standards, as well as rigorously detailing procedures.

Validity of the findings

The research is useful for the scientific area, providing support for this scale’s psychometric properties, distinguishing uses of the scale in its two versions, and arguing well for the added value of the First-Order Four Factor Model. Moreover, it also provides support for its use among students with and without NSSI through measurement invariance, finding that students with NSSI show lower levels in specific subscales, which can be valuable for future research. Conclusions are consistent with the aims proposed and results obtained, acknowledging limitations and how future research could provide further support for hypothesis (e.g. assessing measurement invariance in clinical samples (lines 348 – 352).

Additional comments

This article is an example of a sound, interesting and useful piece of research and we suggest acceptance.

---

## Round 0.2 · accepted · Accept

Thank you for making the requested changes.

Reviewer 1 ·

Basic reporting

no comment

Experimental design

no comment

Validity of the findings

no comment

Additional comments

The authors have addressed each comment by providing sufficient rebuttals or revisions.

Reviewer 2 ·

Basic reporting

All of my comments have been adequately addressed.

Experimental design

All of my comments have been adequately addressed.

Validity of the findings

All of my comments have been adequately addressed.

Additional comments

All of my comments have been adequately addressed.